# Rapid Automatized Naming, Verbal Working Memory, and Rhythm Discrimination as Predictors of Reading in Italian Undergraduate Students with and without Dyslexia

**DOI:** 10.3390/brainsci8050087

**Published:** 2018-05-13

**Authors:** Alice Cancer, Alessandro Antonietti

**Affiliations:** Department of Psychology, Catholic University of the Sacred Heart, 20123 Milan, Italy; alessandro.antonietti@unicatt.it

**Keywords:** developmental dyslexia, reading, rapid automatized naming, rhythm, verbal working memory, undergraduates

## Abstract

Whereas the clinical manifestations and the neuropsychological predictors of Developmental Dyslexia (DD) are already well documented in Italian-speaking children, empirical evidence on DD in Italian adults is in contrast rather scarce. The aim of the present study was to investigate the role of a subset of neuropsychological skills, which have been identified by previous literature to be related to reading, in the decoding abilities of a group of Italian undergraduates with and without DD. For this purpose, 39 university students aged between 19 and 27 years, 19 of whom with a diagnosis of DD, underwent an assessment battery including standardized reading tests, rapid automatized naming (RAN), verbal working memory, and rhythmic pattern discrimination tests. Cross-group differences confirmed significantly lower performances of undergraduates with DD in all measures but rhythm discrimination, compared to typical readers, thus showing a non-compensated neuropsychological profile. Regression analyses showed that, while reading speed was strongly and uniquely predicted by RAN speed, reading accuracy was concurrently predicted by RAN and rhythmic abilities. Finally, RAN speed emerged as a strong predictor of reading performance and risk of receiving a diagnosis of DD.

## 1. Introduction

The difficulties associated with developmental dyslexia (DD) persist into adulthood [1,2,3]. Even adults with DD—who, thanks to compensatory mechanisms, manage to bridge the gap between their prose reading performance and that of typical readers—continue to exhibit impairments in several reading-related abilities, such as phonological processing [4,5], pseudo-word reading [4], and spelling [2].

Although DD has been widely investigated in childhood and adolescence, research on DD in adults is yet rather scarce. The need for a better understanding of the neuropsychological and behavioral manifestations of DD in older populations is fueled by the recently growing number of students with a diagnosis of DD undertaking university studies [6]. In Italy, an exponential growth of +108% of students with a diagnosis of learning disability (LD) who enrolled in University courses was observed from 2012 to 2015 (data analyses by CENSIS (Italian national center for social research); data collection by MIUR (Italian ministry of education), 2017). Further knowledge on the specific markers of Italian adult dyslexia would foster the development of teaching and learning practices specifically designed for this population.

Due to its multidimensional clinical manifestation, a multifactorial neurocognitive explanatory model has been applied to DD [7]. Evidence from developmental studies showed, along with the decoding impairment, concurrent deficiencies in many reading-related cognitive abilities. Among these, phonological awareness has been reported to be the strongest predictor of reading ability in DD—for a review, see [8]—together with rapid automatized naming (RAN) [9,10,11], namely, the ability to rapidly and sequentially name familiar visual stimuli (such as colors, objects, or digits).

A large-scale cross-linguistic study showed that predictors of DD vary across languages, due to differences in orthographic depth [12]. More precisely, the predictive power of phoneme awareness and RAN increased with orthographic complexity, based on the consistencies of phoneme–grapheme correspondences, which differ across languages. In contrast, verbal short-term and working memory (WM) were found to have a significant but comparatively minor impact, with their prediction being, however, more consistent across languages [12].

The relationship between RAN and reading supports the notion that RAN accounts for variance in reading beyond the effect of phonological awareness and verbal short-term memory; e.g., [9,13,14]. Moreover, it was found that the impact of RAN changes across development and type of reading outcomes (speed vs. accuracy). Whereas the correlation between RAN and reading speed remains stable over time, the correlation with reading accuracy tends to decline over school years [15,16]. 

A double-deficit hypothesis of DD [10,17] has been posited on the evidence of RAN and phonological awareness independent contribution to reading problems. The authors of such a hypothesis identified three DD subtypes according to the specific impairment in either one or both processes: readers who show phonological deficits and average naming speed; readers with naming speed deficits and average phonological skills; and readers with a double deficit, who are likely to have the most severe manifestation of DD. However, contrasting findings failed to provide support for the existence of all three deficit subtypes of DD in either children [11] or adults [18], thus questioning the validity of the double-deficit framework. Furthermore, the independence of rapid-naming from phonological processing, based on the evidence of their unique contribution to reading, has been argued by Ziegler and colleagues [19], who pointed to the large variance that the combination of RAN and phonological awareness share in predicting reading [20]. The authors further suggested that, when measures of phonological awareness are not sensitive enough or reach the ceiling (as in the case of transparent languages), most of the shared variance would be left to RAN, which thus would become the key predictor.

As for WM, links between reading performances of children with DD and WM, measured by span tasks, are well established [21,22,23]. The nature of this relation was interpreted as the consequence of a deficit in the phonological loop component of WM, affecting both verbal storage and more complex activities combining storage with processing [24].

Along with the core predictors of DD, a more basic ability has been observed to be associated with dyslexia-related difficulties, namely, auditory temporal processing [25,26,27]. Within the Temporal Sampling framework [28,29], a disruption in the ability to process the fine-grained temporal structure of acoustic stimuli has been suggested to hinder the development of the phonological skills critical for language and reading acquisition in individuals with DD. The rhythmic perceptual ability—measured by same-different judgement tasks in which individuals are required to discriminate between sequences of notes with modulated metrical arrangements—was found to be a strong predictor of word reading in both typical readers and children with DD [27]. Furthermore, the accurate perception of rhythmic structures was shown to predict significant unique variance in reading by longitudinal multiple regression analyses even after phonological awareness was accounted for [30]. Consistent evidence was provided by a study with a sample of Italian children with DD, showing meter perception and rhythm processing as significant predictors of text and pseudo-word reading accuracy as well as word reading speed [31].

Although the majority of research on the predictive nature of the aforementioned cognitive skills has primarily focused on children, few contributions reported significantly lower performances in RAN [32], WM [33,34], and auditory temporal processing [34] in adults with DD as well. 

In order to extend the existing research on the predictive role of such a sub-set of reading-related cognitive abilities to older populations and in a language with a regular orthography, the associations between RAN, verbal WM, rhythmic perceptual abilities, and reading performances have been explored in a study involving Italian undergraduate students with and without DD.

Our decision to focus only on RAN, WM, and rhythmic abilities as predictors of reading in an Italian adult population, thus excluding phonological awareness, stemmed from the tendency for readers in a transparent language to score highly in phonological awareness tests [11]. Moreover, it has been shown that, whereas in beginning readers phonological skills contribute strongly to reading, in more experienced readers a gradual shift towards stronger contribution of RAN typically occurs [35].

The main aims of the present study were the following: (a) Investigating the specific reading and neuropsychological profile of a group of Italian undergraduates with a diagnosis of DD. To do so, their performance in a battery of behavioral tests assessing reading and a subset of reading-related abilities—which have been identified as core predictors of DD in transparent orthography by previous literature concerning children—was compared with that of a typically-reading control group; (b) Measuring the independent and combined contributions of rapid-naming, verbal WM, and rhythmic skills on specific reading parameters—namely, reading speed, reading accuracy, and global reading performance—in a sample of undergraduates who are native speakers of a transparent language (i.e., Italian); (c) Testing the contribution of the strongest predictor of reading performance in estimating an Italian undergraduate’s probability of receiving a diagnosis of DD.

## 2. Materials and Methods

### 2.1. Participants

Thirty-nine Italian undergraduate students, aged 19–27 (M = 22.23; SD = 1.65), participated in the study. Participants were enrolled in undergraduate courses (i.e., Psychology, Medicine, Architecture, Design, Economics, Law, Educational Sciences, Statistics, Political and International Sciences, History) at three universities of Milan, Italy. Their average academic grade level at recruitment was of 26.25 (SD = 1.90), considering the 30-point scale used by Italian universities for grading ordinary exams. Since the assessment battery included a measure of rhythm discrimination ability, we controlled for the influence of musical expertise. Hence, we collected information about participants’ music practice: 18 participants reported to have practiced a music instrument (i.e., guitar, bass guitar, piano, drums, flute, violin, cello, clarinet, or ukulele) for at least 1 year.

Nineteen participants out of the whole sample had a diagnosis of DD (ICD-10 code: F81.0), made by independent clinicians prior to the enrolment in the study, on the basis of standard inclusion and exclusion criteria (i.e., IQ ≥ 85, no sensory or neurological dysfunction) (ICD-10: World Health Organization, 1992) and of the diagnosis procedure followed in Italy [36]. All participants with DD completed their clinical assessment within three years prior to entering university, as requested by Italian law on learning disability aids in higher education (legge n.170 del 2010 art. 3; Accordo Stato Regioni del 24 July 2012). Comorbidities with psychiatric or emotional disorders have been excluded, whereas comorbidities with other specific LD (i.e., dysorthography, dysgraphia, dyscalculia) were accepted. Such a sub-sample was recruited through the Learning Disability service of each university, which provides aid, counselling, and tech support for undergraduate students with specific LDs. For those participants, the enrolment in this study was required to test their eligibility to participate in an intervention study, aimed at measuring the efficacy of a novel training method for improving reading [37,38,39]. 

As for the 20 participants without DD, they had no history of learning problems, nor of educational assistance, had not repeated a grade, and were not on medication. By selecting participants in the highest range of education, and controlling for the aforementioned exclusion criteria, we have minimized the risk of recruiting individuals with other uncorrelated deficits. Typically-reading participants were recruited through each professor of the course in which they were enrolled, who asked them to participate as subjects in at least one experimental study out of a list of ongoing scientific projects, in which the present study was included. Participation was on a voluntary basis and no pressure to take part to the study was exerted. Refusal to participate did not reduce the benefits of participants, who could interrupt participation at any phase of the procedure.

Students who agreed to participate were individually contacted by the researcher and, prior to their enrolment in the investigation, written informed consent was obtained. Clinical records and diagnosis documentation of the sub-sample with DD were collected as well.

To exclude the influence of potential confounding factors, the two sub-groups of participants (i.e., typical-reading undergraduates and undergraduates with a diagnosis of DD) were matched for age, academic grade level, gender, and music expertise (see Table 1).

### 2.2. Procedure

Participants were asked to undergo a neuropsychological assessment including Italian tests for measuring reading abilities, RAN, verbal WM (digit span), and rhythmic discrimination ability. The battery of tests was administered by a psychologist specialized in cognitive assessment during individual sessions taking place in a quiet room.

#### 2.2.1. Battery of Tests

Participants’ reading skills and reading-related cognitive abilities were measured using the following battery of tests. 

*Text reading.* The ability to read aloud prose texts was assessed using the ‘Text reading speed test for high-school and higher education’ [40], which comprises two different passages (i.e., ‘Funghi in città’ [Mushrooms in the city] and ‘Viaggio con le mucche’ [A journey with cows], hereafter referred to as FC and VM respectively). The test provides both accuracy scores, expressed in number of reading mistakes, and speed scores, expressed in syllable per second (syll/s).

*Verbal working memory.* Forward and backward digit span subtests were used to assess verbal WM [41]. Scores have been computed as number of correct items in each subtest.

*Rapid Automatized Naming.* Naming of familiar stimuli was assessed using the ‘Rapid Automatized Naming test–RAN’ [42]. Two subtests from the battery were used, namely ‘Colors’ and ‘Figures’. As for the ‘Colors’ subtest, two matrices (10 rows of 5 stimuli each) of 1 × 1 cm colored squares (i.e., black, blue, red, yellow, and green) were presented. As for the ‘Figures’ subtest, two matrices (10 rows of 5 stimuli each) of black and white figures (i.e., pear, train, dog, star, hand) were presented. For both subtests, the participant was asked to sequentially name each visual stimulus of the matrix as quickly and as accurately as possible. Naming speed (expressed in seconds) and naming accuracy (expressed in number of naming errors) were recorded. In the present study, mean raw scores comprising both subtests (‘Colors’ and ‘Figures’) have been computed and used for the analyses.

*Rhythm discrimination.* The ability to discriminate acoustic rhythmic patterns was assessed using an ad hoc devised test with a same-different-judgment task. Twenty pairs of rhythmic patterns of drum same-pitch notes were presented as pre-recorded auditory stimuli. All items featured a complex time signature (i.e., 5/4 time), so as to set a suitable difficulty level for an adult population. Straight after listening to each pair of rhythms, the participant had to judge whether they were identical or they differed in metrical structure. Scores were computed as the number of incorrect responses.

#### 2.2.2. Data Analyses

First, we measured differences in performance between the two sub-groups (dyslexia vs. non-dyslexia) in all tasks using non-parametric tests. In order to investigate further the differences in reading profiles of undergraduates with and without DD in terms of specific decoding parameter development, *z*-scored reading speed and accuracy measures were compared for each subgroup using non-parametric statistics. 

Then, we employed regression analyses to test the predictive role of the abilities in question on reading. More precisely, three linear regression models were tested to measure the contribution of RAN speed, RAN accuracy, verbal WM, and rhythm discrimination in predicting the variance of each reading parameter (i.e., speed vs. accuracy) and global reading performance (computed as composite score of VM and FC text reading speed and accuracy and expressed in *z*-scores).

Afterwards, to investigate whether the strongest predictor of individual differences in global reading performance would produce a good prediction of diagnosis of DD in undergraduates, a logistic regression model was tested using the binary outcome dyslexia/non-dyslexia.

Effected sizes have been reported as *η*^2^. For the non-parametric comparisons, the magnitude of the effect sizes was calculated according to the equation: *η*^2^ = Z^2^/(N − 1) (where Z is the *z*-score of the test statistic and N is the number of observations).

All reported *p*-values are two-tailed.

## 3. Results

### 3.1. Reading, RAN, Working Memory, and Rhythm Discrimination Differences between Undergraduates with and without DD

Mean values of the observed measures have been compared between the two sub-groups of participants, namely, undergraduates with and without DD, prior to testing regression models including reading and diagnosis of DD as outcomes.

Significant differences emerged for almost every measure (see Table 2). More precisely, reading was significantly impaired in the DD group. The mean composite *z*-scored reading measure in the DD group was below the cut-off of −2 *SD*s from the mean, as provided for by the Italian diagnosis criteria [36]. Such results confirmed the accuracy of the LD clinical reports of the participants with DD. Both reading speed and accuracy performances were significantly lower in the DD group, compared to the non-dyslexic group. Such deficiencies in both reading parameters showed the persistency of speed and accuracy problems associated to DD into adulthood.

Both RAN speed and accuracy performances differed significantly between the two groups, with the DD group showing a slower and more inaccurate RAN. As for verbal WM, a significant difference emerged in both forward and backward digit span subtests, confirming difficulties in the DD group. Finally, the two subgroups performed similarly in the rhythm discrimination task, as no significant differences emerged.

### 3.2. Reading Parameters Differences: Speed vs. Accuracy

To investigate the characteristics of reading profiles in adults with a diagnosis of DD, *z*-scored performances in each reading parameter, namely, reading speed and accuracy, were compared considering each subgroup of participants separately.

A significant difference was found between reading speed and reading accuracy in the DD subgroup (T = 31.00; *p* = 0.01; *η*^2^ = 0.17), with reading speed performance being lower than accuracy performance (speed = −5.13 ± 3.89; accuracy = −2.12 ± 1.45). As expected, typical readers showed normative and similar performances in both reading parameters (T = 94.00; ns) (see Figure 1).

### 3.3. Linear Regression Models of Reading in Undergradutes

Three linear regression models were tested to measure the predictors’ contributions to each specific reading parameter (i.e., speed and accuracy) and to global reading performance. To exclude multicollinearity between predictors, correlation matrices have been computed. Significant Pearson’s *r*’s ranged between 0.35 and 0.52; therefore no strong correlations emerged. Furthermore, collinearity diagnostics confirmed the assumption of no multicollinearity in the multiple regression models performed (all VIF values ranged between 1.03 and 1.37).

Firstly, reading speed was considered as the dependent variable. RAN speed only was found to significantly predict reading speed (β = −0.67; *p* < 0.0001), accounting for a large portion of its variance (*R*^2^ = 0.44; *F*_(1,37)_ = 29.46; *p* < 0.0001).

Secondly, a multiple regression model with reading accuracy as the dependent variable was tested. RAN speed, RAN accuracy, verbal WM, and rhythm discrimination were entered as predictors using the backward method. Results showed that verbal WM was not a significant predictor and therefore it was excluded from the model. On the other hand, RAN speed (β = 0.37; *p* < 0.01), RAN accuracy (β = 0.38; *p* < 0.01), and rhythm discrimination (β = 0.28; *p* < 0.05) together were found to significantly predict reading accuracy, accounting for a large portion of its variance (*R*^2^ = 0.53; *F*_(3,35)_ = 13.30; *p* < 0.0001) (see Figure 2).

Finally, text reading composite score (global reading) was tested as the dependent variable. RAN speed (β = −0.71; *p* < 0.0001), RAN accuracy (β = −0.49; *p* < 0.01), and verbal WM (β = 0.42; *p* < 0.01) were found to predict reading performance independently. More precisely, RAN speed accounted for 50% of global reading variance (*R*^2^ = 0.50; *F*_(1,37)_ = 37.11; *p* < 0.0001) and RAN accuracy for roughly 25% (*R*^2^ = 0.24; *F*_(1,37)_ = 11.58; *p* < 0.01), whereas verbal WM contribution was smaller (*R*^2^ = 0.18; *F*_(1,37)_ = 8.15; *p* < 0.01) (see Figure 3).

### 3.4. Predicting Dyslexia in Undergraduates

After the regression analyses, which led us to identify RAN speed as the strongest predictor of global reading performance in undergraduates, we tested if an individual’s probability of receiving a diagnosis of DD changes according to their RAN speed, employing a logistic regression model with the binary outcome dyslexia/non-dyslexia.

Results confirmed that RAN speed produced a strong prediction of diagnosis of DD in undergraduates (Model: χ^2^ = 23.97; *p* < 0.0001). Nagelkerke’s *R*^2^ of 0.61 indicated that RAN speed significantly contributed to the prediction of grouping participants according to their diagnosis of DD (B = 0.24; OR = 1.27; *p* = 0.01) (see Figure 4).

## 4. Discussion

The aim of the present study was to explore whether and to what extent a set of reading-related cognitive abilities—namely, RAN, verbal WM, and rhythm discrimination—predicted reading skills in Italian undergraduate students with and without DD.

Our decision to focus on undergraduate students was driven by the exponentially growing number of Italian individuals with a diagnosis of LD who decide to undertake university studies and enroll in LDs services for receiving special education aids. We suggest that a better understanding of their specific neuropsychological profile, relative to their typically-reading peers, and the identification of the cognitive predictors of their specific reading difficulties would foster the development of diagnostic and rehabilitation tools specifically designed for this population.

The two observed reading parameters (that is, text reading speed and accuracy) were found to be both significantly impaired in the DD group (with both mean *z*-scores exceeding the clinical cut-off of −2 SDs below the normative sample mean, as defined by Italian diagnosis criteria [36]). Furthermore, the reading speed deficit was found to be significantly more severe compared to the accuracy deficit. Such results seem to discard the hypothesis according to which the compensatory cognitive mechanisms of individuals with DD would reduce the performance gap for reading speed over development and would increase that of reading accuracy, relative to typical readers [43]. 

RAN and verbal WM measures were found to be significantly lower in the subgroup of participants with a diagnosis if DD, relative to the typically-reading group, consistently with previous literature on adult DD [32,33,34]. Together with the evidence of impaired reading, such cross-group differences support an overall non-compensated neuropsychological profile in Italian adults with DD, who continue to manifest the specific deficits characterizing DD in the developmental age.

Rhythmic abilities, as measured by the rhythmic discrimination task, did not differ between groups.

As for the unique and concurrent contribution of the studied predictors to reading, RAN speed was found to be a strong independent predictor of reading speed, explaining 44% of its variability. As for reading accuracy, 53% of its variance was shared between RAN speed, RAN accuracy, and rhythm discrimination. Finally, global reading performance was independently predicted by RAN speed, accounting for 50% of its variance alone and, on a smaller scale, by RAN accuracy and verbal WM.

Our finding that RAN is predictive of both text reading speed and accuracy is consistent with the results by Di Filippo and colleagues [9] in typically-reading Italian children. However, such results do not confirm the longitudinal tendency observed in English typically-reading children [15,16] showing a stable correlation between RAN and reading speed over time, with a decline of the correlation with reading accuracy over school years. Consistently with our results, RAN and verbal WM concurrent prediction of text reading performance were reported by Nergård-Nilssen and Hulme [3] in a study with Norwegian adults—individuals speaking a language with a moderately consistent orthography—with DD. We argue that such a relationship between prose reading and rapid-naming can be explained by the fact that the latter is involved when groups of letters or entire words are processed as whole-units, rather than as a sequence of grapheme-phoneme correspondences [13]. Indeed, neurofunctional evidence showed a correlation between the performance in prose reading and the activation of a left ventral occipitotemporal region, which is engaged by visual-orthographic whole word recognition in readers with DD in a regular language (German) [44]. 

The ability to detect differences in auditory rhythmical patterns was found to be associated with reading performances in Italian adults with DD, consistently with many developmental studies [25,26,27], thus supporting the evidence of an auditory temporal processing deficit in DD e.g., [31] and adducing further evidence to the Temporal Sampling framework of DD [28,29].

The contribution of rhythmic abilities to the prediction of text reading was found to be specific for the accuracy parameter, consistently with the findings by Flaugnacco et al. [31], who reported a meter perception task to be a good predictor of text reading accuracy in Italian children with DD. A strong predictive role of both verbal working memory and rhythmic abilities in reading performances that involved a lexical-route activation (i.e., word reading) have been also reported by Fostick and colleagues [34,45] in a study with Hebrew adults with DD. We suggest that the lack of cross-group differences for rhythm discrimination can be explained by the fact that the contribution of rhythmic skills was found in a specific reading parameter only—i.e., accuracy—that was significantly less impaired, relative to reading speed, in the DD group.

Finally, RAN speed produced a strong prediction of the probability to receive a diagnosis of DD, accounting independently for 61% of its total variation. The binary dependent variable entered in the logistic regression model (i.e., dyslexia, non-dyslexia) was defined by the diagnoses of DD made by independent clinicians prior to the enrollment in the present investigation. The standard Italian diagnostic procedure for DD [36] includes, other than a comprehensive reading and intellectual assessment, a clinical judgment based on individual’s medical, developmental, educational, and family history, and exclusion criteria, which could better explained the performance impairments (i.e., variation of academic attainment, neurological or sensory disorders, neurodegenerative cognitive disorders, etc.). Furthermore, such a diagnostic procedure does not include RAN or rhythmic discrimination tests, as is the case of our sample. Our finding that RAN speed alone produced a strong prediction of the probability to receive a diagnosis of DD, accounting for more than 50% of its variance, has a valid clinical implication. For example, the RAN task appears to be a potential reliable screening instrument to detect the risk of DD in adults. A computerized version of the task could be implemented for a quick self-administration, which could be included in a protocol for adult self-report of DD together with other screening tools, such as self-report scales [46], to improve their predictive validity.

The selection of the predictors and outcome measures considered in the present investigation was driven by both methodological and theoretical reasons. As for the predictors, only RAN e.g., [11,32], verbal WM e.g., [23,33], and rhythmic abilities e.g., [27,34] have been measured, on the bases of previous child and adult dyslexia literature. We decided to exclude phonological measures due to the tendency for readers in a transparent language to score highly in phonological awareness tests [11]. Furthermore, a longitudinal tendency from a strong contribution of phonological skills in the beginning phases of learning to read, followed by a gradual shift towards stronger contribution of RAN in more experienced readers have been observed [35]. As for the outcome reading measures, we did not include word and pseudo-word reading due to the lack of Italian normative data for university students. No reading comprehension measures were applied since the comprehension of written language relies on both decoding skills and oral language comprehension skills [47], thus resulting in an opaque measure. Therefore, we decided to focus only on the decoding component of prose reading. 

We exclude that articulation could have had a significant role on reading or RAN performance. In their study on Italian children, Di Filippo and colleagues [9] found no predictive role of articulation rate on reading performance, whereas the association between RAN and reading was confirmed. Furthermore, Georgiou and colleagues [48] recently found that pause time in the RAN task explained more reading speed variance than articulation time in transparent orthographies, such as Greek and Finnish. 

As any small-scale investigation, the present study has a few limitations that should be taken into account, the first being the small sample size (*N* = 39). However, with a sample size of 39, the statistical power of a multiple regression model comprising 4 predictors would be 0.99, with a large effect size of (*R*^2^ = 0.50; 𝛼 = 0.05). Power analyses for the regression models applied to our data ranged from 0.81 to 0.99. We therefore assumed that, in spite of being rather small, a sample size of 39 would be adequate for employing the regression analyses here reported.

Another potential limitation of the study is the different methods of recruitment used for the undergraduates with and without DD. We understand that undergoing a neuropsychological assessment could potentially have a different meaning for the participants of two sub-groups. We argue that, since participants of both groups had, however different, a motivation to participate in the study, no major difference in the level of engagement would emerge. 

## 5. Conclusions

The aim of the present study was to extend the knowledge about the predictive nature of a set of cognitive skills, widely investigated in child populations, to Italian adults’ reading ability. RAN, verbal WM, and rhythmic perception abilities were found to be associated with prose reading speed and accuracy in a sample of undergraduate students with and without a diagnosis of DD. 

Both reading speed and accuracy performances were found to be significantly impaired in the subgroup of participants with a diagnosis of DD, with the speed parameter being significantly lower compared to accuracy. It has been previously reported that, in individuals with DD, compensatory mechanisms would contribute over development to overcome the reading performance deficit, even though mostly phonological processing and spelling deficits would persist into adulthood [32,49]. Our findings showed that, in Italian undergraduates with DD, prose reading performances are still significantly lower compared to typical readers, together with lower RAN and verbal WM skills. Furthermore, it has been suggested that the reading performance trajectory pattern of Italian individuals with DD over time would reduce the distance from typically-reading peers for speed, while increasing that for accuracy [45]. Our finding did not confirm such a developmental trend, since both accuracy and speed parameters were found to be significantly impaired in adults with DD, with speed reading being more severely deficient. Whereas reading speed was found to be strongly and uniquely predicted by RAN speed, reading accuracy was concurrently predicted by RAN and rhythmic abilities. Finally, RAN speed emerged as a strong predictor of reading performance and risk of DD in adults. 

Such findings contribute to extend the knowledge about the development of a neuropsychological profile of adults with DD and may have some useful clinical implications. The predictive contribution of RAN could be exploited by devising a screening instrument for DD in adults including RAN tasks. Identifying DD in adults is indeed important for supporting those who want to undertake university studies, whose number is growing [6], and thus may be eligible to receive the dedicated aids for LD university students, for instance by exploiting the creative potential showed by individuals with DD [50,51]. 

Finally, the dyslexia-related problems that have been found to persist into adulthood may be targeted by remedial interventions specific for adults with DD. As an example, the association between reading accuracy and rhythmic skills may suggest to take benefit from rhythmic interventions specifically focused on synchronization mechanisms in reading, which have been proved to be effective also in undergraduates with DD [37].

## Figures and Tables

**Figure 1 brainsci-08-00087-f001:**
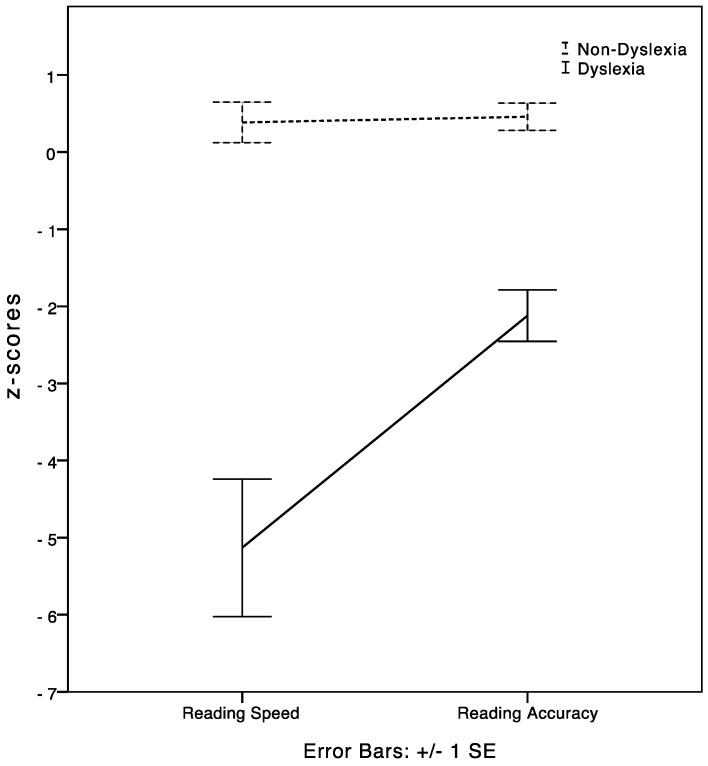
Text reading speed vs. text reading accuracy *z*-scores differences in the two subgroups of participants (i.e., undergraduates with and without DD).

**Figure 2 brainsci-08-00087-f002:**
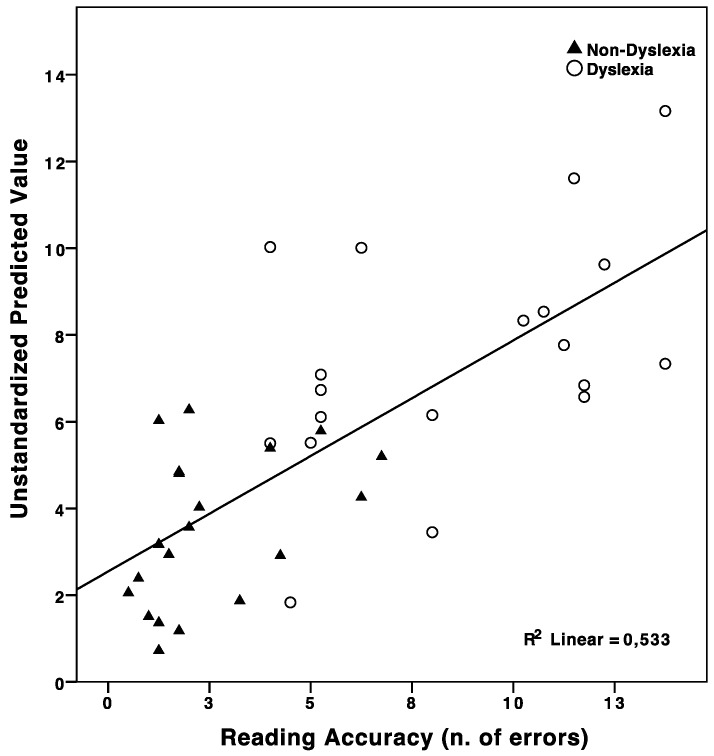
Scatterplot depicting the predicted values from the multiple regression equation (Rapid automatized naming (RAN) speed, RAN accuracy and rhythm discrimination) vs. the dependent variable (reading accuracy). Cases are labelled according to DD diagnosis.

**Figure 3 brainsci-08-00087-f003:**
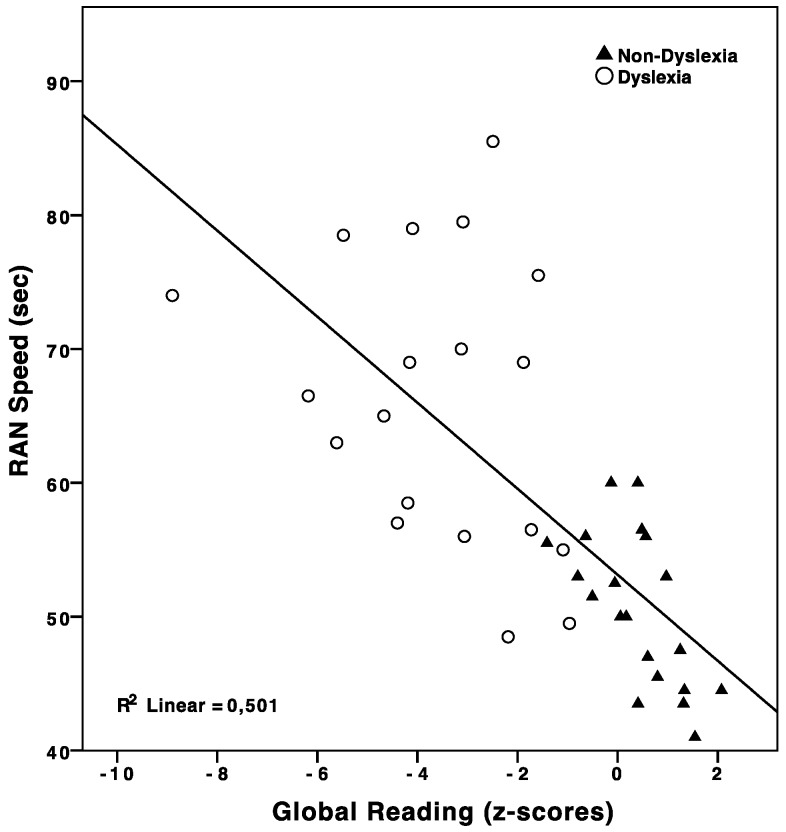
Scatter plot reading vs. RAN speed. Cases are labelled according to DD diagnosis.

**Figure 4 brainsci-08-00087-f004:**
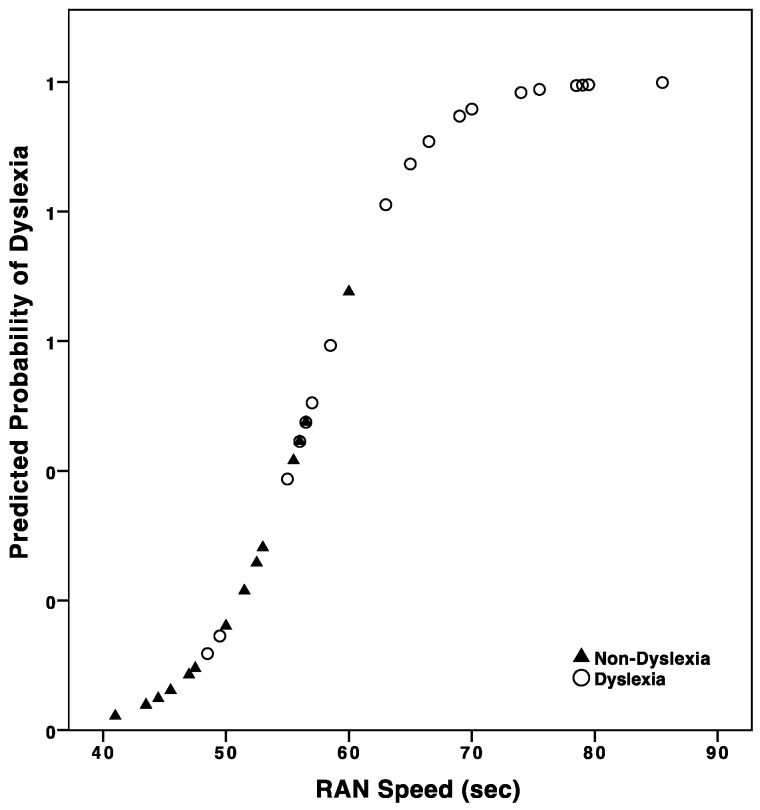
Probability curve showing the change in risk of dyslexia by RAN speed, extracted from the logistic regression model.

**Table 1 brainsci-08-00087-t001:** Participants’ characteristics and comparisons of age, academic grade level, gender, and music expertise between undergraduates with and without developmental dyslexia (DD).

Mean (SD), resp. N (%)	Non-Dyslexia *N* = 20	Dyslexia *N* = 19	Comparisons ^1^	
			χ^2^, *p*	*t*_(37)_, *p*
Age	22.69 (0.98)	21.74 (2.05)	-	1.82, 0.08
Academic Grade	26.36 (1.23)	25.43 (2.17)	-	1.96, 0.06
Males	8 (40%)	10 (52.6%)	0.63, 0.53	-
Music Experts	12 (60%)	9 (47.4%)	0.63, 0.53	-
Years of Music Practice	3.70 (5.98)	2.06 (2.73)	-	1.25, 0.22

^1^ Means were compared using the independent *t*-test; percentages for categorical data were assessed by χ^2^ test.

**Table 2 brainsci-08-00087-t002:** Comparisons of reading and reading-related performances between undergraduates with and without DD. Ranked positions of scores were compared using the Mann–Whitney *U* test. Effect sizes are reported as *η*^2^; * *p* < 0.05; ** *p* < 0.01; *** *p* < 0.0001.

Mean (SD)	Non-Dyslexia *N* = 20	Dyslexia *N* = 19	Comparisons
			*U*, *η*^2^
**Global Reading** ^1^	0.42 (0.86)	–3.62 (2.01)	2.00 ***, 0.73
**Read. Speed** ^2^	6.12 (0.96)	3.59 (1.34)	12.00 ***, 0.66
**Read. Accuracy** ^3^	2.50 (1.84)	8.55 (3.50)	24.00 ***, 0.57
**RAN Speed** ^4^	50.55 (5.74)	66.08 (10.83)	40.50 ***, 0.46
**RAN Accuracy** ^3^	0.97 (0.94)	2.74 (2.47)	96.50 **, 0.19
**FWD Digit Span**	8.25 (2.17)	5.84 (1.57)	72.00 **, 0.30
**BKD Digit Span**	7.40 (1.82)	6.26 (1.19)	118.50 *, 0.11
**Rhythm Discr.** ^3^	2.70 (2.25)	4.05 (2.86)	136.00, 0.06

^1^ Composite score of VM and FC text reading speed and accuracy, expressed in *z*-scores. Negative values correspond to a performance below the population’s mean; ^2^ Raw speed scores expressed in syll/s, thus higher values correspond to a better performance; ^3^ Raw accuracy scores expressed in n. of errors, thus lower values correspond to a better performance; ^4^ Raw speed scores expressed in seconds, thus lower values correspond to a better performance.

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
