# Peer review of "Rapid Automatized Naming, Verbal Working Memory, and Rhythm Discrimination as Predictors of Reading in Italian Undergraduate Students with and without Dyslexia"

_brainsci, 2018, doi:10.3390/brainsci8050087_

Round 1

Reviewer 1 Report

Brain Sciences – 304248 R

The authors examined both the reading (speed, accuracy) and reading-related skills (RAN, Working memory, rhythmic discrimination) of dyslexic vs. matched neurotypical adults, focusing on the (1) correlations between dependent variables (reading and reading-related skills), (2) cross-group differences, (3) the predictive strength of reading-related skills regarding reading performance, and (4) the potential of RAN speed to predict a diagnosis of Developmental Dyslexia.

A) While the current version of the introduction seems more consistent than the previous, unrevised version, I still see some issues to be solved:

1-One relates to the motivation and meaning of the four levels of analysis, and to how the corresponding four levels of results are integrated in the discussion. To me, the purpose of levels (3) – what predicts reading skills in adult DD - and (4) – does RAN speed alone predict a diagnosis of DD – is clear. However, in the overall context, the same does not go for levels (1) – correlations among all dependent variables- and (2) cross-group differences.

1.1  What do these levels (1 and 2) add to the main findings presented in the abstract – which focuses on levels (3) and (4)? Moreover, how do all levels fit together? For instance, the authors found no cross-group differences for rhythmic discrimination, but they did find the latter to predict reading accuracy.

1.2 My suggestion would be either (a) to delete levels (1) and (2), or to make clear why these analyses are necessary and how they integrate with the remaining (main) findings.

2- The rhythmic task was a discrimination task. The authors used accuracy (its counterpart – number of errors) as a dependent variable. However, it is know that global accuracy measures can be misleading in discrimination tasks (the amount of errors could be 50% simply because participants decided to press always the same key and if there are half different, half same). Discrimination measures (d-prime) would be necessary for a proper assessment of participants’ performance. I wonder if the authors tried those.

3-In case the authors do believe that correlations are a necessary piece in the analysis (and present reasons for that), I would like to ask whether the authors have done any kind of p-value correction for multiple correlations.

3.1. To me, one obvious role of the correlations should be demonstrating lack of multicollinearity among reading-related predictors (a pre-requisite for a reliable regression procedure), but that is never mentioned.

B) In addition, I see minor style issues, which I point below:

P1, ln 30: comma after typical readers

P2, ln 78-81: The constituent “Within the temporal sampling framework” should go to the beginning of the sentence

P2, ln 87-89: comma after …”with DD”;

Should also read “…pseudo-worD reading accuracy AS WELL AS…”

P3, ln 104: followinG (not followings)

P3, ln 107: M and SD italicized

P3, ln 121: the reference to a 30-point scale for grade level requires further explanation (the tendency is to see the number as referring to years of schooling, but that would make no sense)

P3, ln 139: “or” instead of “nor”, I think

P4, ln 146: the phrasing is awkward. I would say “Refusal to participate did not reduce the benefits of participants…”

P4, ln 147: AT any phase

P4, ln 185: I would say PAIRS instead of couples

P5, ln 188: I would say PAIRS OF RHYTHMS instead of couple of acoustic patterns (everything in the tests consists of acoustic patterns, this is too vague)

P5, ln 192: I would place the data analysis subsection in the methods section, but I don’t know if this is the journal guideline for that

P5, ln 226: as PREDICTED

P7, ln 278: remove “the” before typically-reading participant; it is also better to use TYPICAL READERS

P10, ln 343-346: switch the order of constituents: according to which the compensatory mechanisms of individuals with DD would reduce the performance gap for reading speed over development, and would…

P10, ln 347: strongly AND positively

P11, ln 362: DO not confirm

P13, ln 456: creative POTENTIAL shown by INDIVIDUALS

Author Response

Reply to the Review Report (Reviewer 2):

A) While the current version of the introduction seems more consistent than the previous, unrevised version, I still see some issues to be solved:

1-One relates to the motivation and meaning of the four levels of analysis, and to how the corresponding four levels of results are integrated in the discussion. To me, the purpose of levels (3) – what predicts reading skills in adult DD - and (4) – does RAN speed alone predict a diagnosis of DD – is clear. However, in the overall context, the same does not go for levels (1) – correlations among all dependent variables- and (2) cross-group differences.

1.1  What do these levels (1 and 2) add to the main findings presented in the abstract – which focuses on levels (3) and (4)? Moreover, how do all levels fit together? For instance, the authors found no cross-group differences for rhythmic discrimination, but they did find the latter to predict reading accuracy.

1.2 My suggestion would be either (a) to delete levels (1) and (2), or to make clear why these analyses are necessary and how they integrate with the remaining (main) findings.

As regards the correlation analyses among all measures (level 1), even though correlations matrices were reported in similar works investigating reading predictors, we followed the reviewer’s suggestion and they are no longer included in the manuscript.

The importance of cross-group differences (level 2) has been now better explained in the light of a comprehensive understanding of the compensatory mechanisms effects, which are frequently described in adult dyslexia literature (P 9, ln 412-415). Furthermore, results of such comparisons have been now included in the abstract. 

It has been now suggested in the discussion (P 10, ln 462-465) that the lack of cross-group differences for rhythm discrimination can be explained by the fact that the contribution of rhythm was found in a specific reading parameter only – i.e., accuracy – which was less impaired in the DD group, relative to reading speed.

2- The rhythmic task was a discrimination task. The authors used accuracy (its counterpart – number of errors) as a dependent variable. However, it is know that global accuracy measures can be misleading in discrimination tasks (the amount of errors could be 50% simply because participants decided to press always the same key and if there are half different, half same). Discrimination measures (d-prime) would be necessary for a proper assessment of participants’ performance. I wonder if the authors tried those.

We thank the reviewer for the suggestion to use sensitivity d’ scores instead of global accuracy measures for the rhythm discrimination task. Our decision to use accuracy scores instead of d-prime scoreswas based on other authors’ works addressing similar topics – namely, the relationship between rhythm and dyslexia –, in which the performance in meter/rhythm discrimination tasks similar to ours was measured considering the number of correct responses (see Flaugnacco et al., 2014, Huss et al., 2011). However, we computed d’ sensitivity scores in our sample as follows: d' = z(H) - z(F) (Macmillan & Creelman, 2004). We found a correlation of –.97 (p < .0001) between d-prime and global accuracy (expressed in number of errors) scores. Therefore, we argue that the influence of response bias was irrelevant in our sample. In fact, when using d’ scores, we found similar results: Rhythm discrimination did not differ between groups (U= 126; n.s.) and was mildly and positively associated with forward digit span (r = .36; p < .05), backward digit span (r = .44; p < .01), text reading accuracy (r = –.34; p < .05). For all these reasons, we decided to maintain global accuracy scores instead of d’ scores for measuring rhythm discrimination.

3-In case the authors do believe that correlations are a necessary piece in the analysis (and present reasons for that), I would like to ask whether the authors have done any kind of p-value correction for multiple correlations.

Correlation analyses have been now excluded from the manuscript.

3.1. To me, one obvious role of the correlations should be demonstrating lack of multicollinearity among reading-related predictors (a pre-requisite for a reliable regression procedure), but that is never mentioned.

To prove the absence of multicollinearity, correlations between predictors’ coefficient range have been reported in the manuscript (all Pearson’s r values < .80). Furthermore, VIF values range for the multiple regression models have been reported as well (P 7, ln 346-350).

B) In addition, I see minor style issues, which I point below:

P1, ln 30: comma after typical readers

P2, ln 78-81: The constituent “Within the temporal sampling framework” should go to the beginning of the sentence

P2, ln 87-89: comma after …”with DD”;

Should also read “…pseudo-worD reading accuracy AS WELL AS…”

P3, ln 104: followinG (not followings)

P3, ln 107: M and SD italicized

P3, ln 121: the reference to a 30-point scale for grade level requires further explanation (the tendency is to see the number as referring to years of schooling, but that would make no sense)

P3, ln 139: “or” instead of “nor”, I think

P4, ln 146: the phrasing is awkward. I would say “Refusal to participate did not reduce the benefits of participants…”

P4, ln 147: AT any phase

P4, ln 185: I would say PAIRS instead of couples

P5, ln 188: I would say PAIRS OF RHYTHMS instead of couple of acoustic patterns (everything in the tests consists of acoustic patterns, this is too vague)

P5, ln 192: I would place the data analysis subsection in the methods section, but I don’t know if this is the journal guideline for that

P5, ln 226: as PREDICTED

P7, ln 278: remove “the” before typically-reading participant; it is also better to use TYPICAL READERS

P10, ln 343-346: switch the order of constituents: according to which the compensatory mechanisms of individuals with DD would reduce the performance gap for reading speed over development, and would…

P10, ln 347: strongly AND positively

P11, ln 362: DO not confirm

P13, ln 456: creative POTENTIAL shown by INDIVIDUALS

We thank the reviewer for all the style and grammar suggestions. Changes have been now made accordingly in the manuscript.

Reviewer 2 Report

This is a revised ms that has adequately addressed reviewer comments, in my opinion.

Author Response

This is a revised ms that has adequately addressed reviewer comments, in my opinion.

We thank the reviewer for his positive comments.